# First Record of *Flavocillium subprimulinum* (Cordycipitaceae, Hypocreales) in Mexico: Morphological and Molecular Characterisation, Nematocidal Activity of Its Liquid Culture Filtrates against *Haemonchus contortus* and Protease Activity

**DOI:** 10.3390/jof10010056

**Published:** 2024-01-09

**Authors:** Gustavo Pérez-Anzúrez, Pedro Mendoza-de Gives, Agustín Olmedo-Juárez, María Eugenia López-Arellano, Génesis Andrea Bautista-García, Ana Yuridia Ocampo-Gutiérrez, Elke von Son-de Fernex, Miguel Ángel Alonso-Díaz, Edgar Jesús Delgado-Núñez, Adolfo Paz-Silva

**Affiliations:** 1Laboratory of Helminthology, National Centre for Disciplinary Research in Animal Health and Innocuity (CENID-SAI), National Institute for Research in Forestry, Agriculture and Livestock-Ministry of Agriculture and Rural Development, (INIFAP-SADER), Jiutepec 62550, Mexico; tavopzaz@gmail.com (G.P.-A.); aolmedoj@gmail.com (A.O.-J.); bagg150583@gmail.com (G.A.B.-G.);; 2Production Sciences and Animal Health, Faculty of Veterinary Medicine and Zootechnics, National Autonomous University of Mexico, Coyoacán, Ciudad de México 04510, Mexico; 3Tropical Livestock Center, Faculty of Veterinary Medicine and Zootechnics, National Autonomous University of Mexico, Martínez de la Torre 93600, Mexico; elkevsf@hotmail.com (E.v.S.-d.F.); alonsodma@hotmail.com (M.Á.A.-D.); 4Faculty of Agricultural, Livestock and Environmental Sciences, Autonomous University of the State of Guerrero, Iguala de la Independencia 40040, Mexico; 5Department of Animal Pathology, Faculty of Veterinary, University of Santiago de Compostela, 27142 Lugo, Spain; adolfo.paz@usc.es

**Keywords:** *Flavocillium*, hypocreales, nematocidal, *Haemonchus*, Mexico

## Abstract

This is the first record of the fungus *Flavocillium subprimulinum* in Mexico. The isolate was taxonomically characterised and cultured in potato dextrose broth (PDB), Czapek–Dox broth (CzDoxB), and sweet potato dextrose broth (SPDB) to obtain its filtrates (FLCF). The nematocidal activity (NA) of three FLCF concentrations was assessed against *Haemonchus contortus* L3. Protease activity (PA) was assessed with SDS-PAGE, followed by a zymogram. The NA of the FLCF reached 94.43% in PDB and 95.82% in CzDoxB, respectively, at 100 mg/mL. Lower mortality (64%) was found in SPDB at 100 mg/mL. SDS-PAGE showed bands (in PBS) of ~25, ~40, and ~55 kDa. The zymogram showed protein bands (PBs) with PA in the media, including PBs of ~14, ~40, and ~55 kDa. This study establishes the basis for exploring the potential use of this fungus against *H*. *contortus*, which is considered the most pathogenic parasite affecting lambs.

## 1. Introduction

Soil mycobiota plays a crucial role in a number of ecological functions, mainly regulating various soil physiological processes. For example, they enhance soil formation, decompose organic matter [1], promote plant growth [2], activate plant defences [3], act as natural biocontrol agents [4], and provide suitable conditions for the establishment of other organisms to maintain soil fertility [5,6]. Fungi belonging to the order Hypocreales are members of the soil mycobiota that are associated with important entomopathogenic activity. In this order, the family Cordycipitaceae has existed since the Cretaceous period [7], with important ecological roles such as acting as biocontrol agents and enhancing plant growth [8]. One of the most widely studied fungal genera belonging to this class is *Lecanicillium*. Species classified into this genus are potential biocontrol agents with important lethal activity against aphids, scales, whiteflies, thrips, mealy bugs, nematodes, and other insect pests [9,10,11]. *Lecanicillium subprimulinum* was reported by Huang et al. as a novel species from Baoshan, Yunnan, China [12]. It was characterised by the presence of conidiophores arising from hyaline hyphae, with gregarious, ellipsoid to ovoid, aseptate conidia. In 2020, a multi-gene phylogenetic analysis of the family Cordycipitaceae was performed, and after the analysis of molecular phylogenetic and morphological data, two new genera (*Flavocillium* and *Liangia*) were introduced. In the case of *Flavocillium*, one new species, *F*. *bifurcatum*, and three combinations that were previously recorded as belonging to the *Lecanicillium* genus, whose species were proposed as *F*. *acerosium*, *F*. *primulinum,* and *F*. *subprimulinum,* were introduced. Of these, the latter is a species characterised by the presence of yellowish stromata [13].

The pathogenic effect of nematophagous fungi against nematodes has been identified using different biological strategies, such as mechanical activity through the production of trapping devices or simple adhesive hyphae [14]. This process seems to be regulated by multiple signaling hubs, and, recently, the conserved cAMP-PKA-signaling pathway has been shown to be involved in the trap formation in the nematode-trapping fungus *Arthrobotrys oligospora*, which is induced by the presence of nematodes [15]. However, a biochemical fungal strategy through secondary metabolites and cuticle-degrading enzyme proteins that contribute to the capture, penetration, and tissue degradation of prey nematodes has been identified [16]. Enzymes produced by members of the Orbiliaceae group, e.g., proteases, serine proteases, and chitinases, have been investigated, particularly in those belonging to *A*. *oligospora*, *Duddingtonia flagrans,* and *Monacrosporium thaumasium*, with an emphasis on their use and activity against nematodes [17]. Thus far, there is little information about protease enzyme activity in members of the Cordicipitaceae family, and most studies have focused on enzymes produced by species of the *Lecanicillium* genus. This genus is phylogenetically related to *Flavocillium* [13,18], and it has been shown to possess two modes of action against the eggs of the root-knot nematode *Meloidogyne javanica*. Hyperparasitism of this fungus against the eggs of *M*. *javanica* has been reported, with hyphae penetrating and destroying the eggshell and feeding on the embryo. Additionally, liquid culture filtrates (LCF) of *Lecanicillium* spp. inhibit egg hatching and cause juvenile destruction in *M*. *javanica*. This lethal strategy has been attributed to the production of lytic enzymes that act in signaling and stress-response functions, bioenergy, metabolism and protein synthesis, and degradation [10].

*Haemonchus contortus* is one of the main pathogenic parasitic nematodes living in the stomachs of sheep and goats, severely affecting their health and productivity [19]. This parasite possesses an oral lancet to puncture the veins of the abomasum (stomach) and sucks blood, causing anaemia, loss of appetite, malnutrition, starvation, and susceptibility to other diseases, even resulting in the death of young animals [20].

To our knowledge, there have been no reports on the biological activity of *F*. *subprimulinum*. In this context, the objective of the present study was to isolate, identify, and assess the in vitro nematocidal activity of the LCF of a Mexican isolate of the fungus *F*. *subprimulinum* against infective larvae of the blood-feeding nematode parasite *Haemonchus contortus* and to identify potential protease activity in the LCF of *F*. *subprimulinum*.

## 2. Materials and Methods

### 2.1. Location

This study was performed at the Laboratory of Helminthology of the National Center for Disciplinary Research in Animal Health and Innocuity of INIFAP-Mexico, Jiutepec City, Morelos, Mexico.

### 2.2. Biological Material

The fungal isolate was obtained from agricultural soil in the Municipality of Tetela del Volcán, Morelos State, Mexico (Latitude 18°53′35.99 N; Longitude 98°43′45.00 W [21] [Figure 1]). Soil samples were collected from an agricultural field and processed following the sprinkling method described by Barron [14], which consists of sprinkling approximately 0.3 to 0.5 g of soil on plates containing water agar and baiting with drops of an aqueous suspension containing a non-determined amount of the free-living nematode *Panagrellus redivivus*. The plates were incubated at room temperature (18–25 °C) for 10–25 days. From the first week, the surface of each plate was observed under a stereomicroscope (at 5×) and a light microscope (at 40 and 75×). Once larvae appeared trapped, the aerial structures were visualised on the agar surface, and monoconidial passes to sterile water agar plates were conducted. This process was performed several times until hyphae and spores were obtained in a pure culture [22].

### 2.3. Fungal Traditional Taxonomy (Morphometrics)

Once the fungus was obtained in pure culture, the most important taxonomic characteristics, including macroscopic and microscopic characteristics, of the colony growing on the agar were analysed. The micro-culture technique was performed to observe the structures in more detail; cotton blue staining was used to contrast the fungal structures. Macroconidia and microconidia shape and size, including length and width, were observed, and the presence or absence of septa in conidia, along with the number of phialides, was determined. A total of 50 structures (either macroconidia and microconidia or phialides) were observed and measured under a microscope to obtain the means of these measurements. These characteristics were compared with those published by Huang et al. [12] and Wang et al. [13] to establish the proposed classification of the genus and species.

### 2.4. Fungal Taxonomy by Molecular Techniques

Mycelia were obtained by cultivating the fungus on potato dextrose broth (PDB) (Merck KGaA, Darmstadt, Germany) in flasks for 15 days at room temperature (18–25 °C). The fungal DNA was obtained using the Wizard^®^ Genomic DNA Purification Kit (Promega, Madison, WI, USA). An IMPLEN spectrophotometer (NanoPhotometer NP80, Munich, Germany) was used to quantify genomic DNA. Subsequently, the ITS-4 (5′-GGAAGTAAAAGTCGTAACAAGG-3′) and ITS-5 (5′-TCCTCCGCTTATTGATATGC-3′) oligonucleotides were used to amplify the DNA ITS region [23], and the EF-1018F (GAYTTCATCAAGAACATGAT) and EF-1620R (GACGTTGAADCCRACRTTGTC) oligonucleotides were used for the TEF1-α region [24] by PCR, using a C1000 Touch Thermal Cycler (Bio-Rad, Hercules, CA, USA). The PCR procedure was performed under the following conditions: for ITS, initial denaturation at 94 °C for 3 min, 35 cycles of denaturation at 94 °C for 1 min, annealing at 42 °C for 90 s, and extension at 72 °C for 90 s, followed by a final extension stage at 72 °C for 5 min; and for TEF1-α, initial denaturation at 95 °C for 5 min, 35 cycles of denaturation at 95 °C for 1 min, annealing at 51 °C for 60 s, and extension at 72 °C for 120 s, followed by a final extension stage at 72 °C for 10 min. The amplicon size was confirmed by electrophoresis on 1.5% agarose gel, and the Wizard^®^ SV Gel and PCR Clean-Up System (Promega, Madison, WI, USA) was employed to purify the obtained products according to the manufacturer´s directions. The amplified regions were sequenced at the Institute of Biotechnology of the National University Autonomous of Mexico (IBT-UNAM) using an Applied Biosystem Sequencer (Thermo Fisher Scientific, Waltham, MA, USA). Sequence alignment was performed using the Basic Alignment Search Tool, BLAST (https://blast.ncbi.nlm.nih.gov/Blast.cgi, accessed on 23 August 2022).

### 2.5. Phylogenetic Tree

The phylogenetic analysis was performed by aligning sequences of the ITS and TEF1- α regions from our isolate with respect to sequences reported in the NCBI database using the BLAST Tool. Forty sequences with the highest similarities with both regions (ITS and TEF1-α) were selected. Multiple alignment was performed using the CLUSTAL 2.0 algorithm in MEGA software (v11.0.13). Both BioEdit (v7.2.5) and MEGA software were used to link the two alignments. The model TIM2 + G was identified as the best substitution model, obtained under the Akaike Information Criterion (AIC) using jModelTest Software (v2.1.10.). Bayesian inference analysis was performed for phylogenetic tree construction using the BEAST^®^ package (v2.7.6) for Windows^®^ (v22H2). The first 10% of the trees were discarded, and the remaining 90% were used to estimate the posterior probabilities of the consensus tree. FigTree Software (v1.4.4) was employed to visualise and edit the obtained tree.

### 2.6. Nematodes

#### 2.6.1. Panagrellus Redivivus

A strain of the free-living nematode *P*. *redivivus* was cultivated in crystal containers (10 cm width × 20 cm height) using oat grains and water, following the procedure described by de Lara et al. [25]. Nematodes were recovered from cultures using the Baermann funnel technique and passed through 74 µm sieves to separate the oat residues. Finally, the nematodes were rinsed in distilled water.

#### 2.6.2. *Haemonchus contortus* Infective Larvae

A population of infective larvae (L3) of the parasite was obtained from faeces of an *H*. *contortus* egg-donor lamb artificially infected with the parasite and maintained under confinement in a pen in the flock experimental area of CENID-SAI, INIFAP. Faeces containing the eggs of the parasite were collected directly from the rectum of the animal. The Norma Oficial Mexicana (Official Mexican Standard) with the official rule number NOM-052-ZOO-1995 (http://www.senasica.gob.mx, accessed on 8 August 2023) [26], as well as the Ley Federal de Sanidad Animal (Federal Law for Animal Health) DOF 07-06-2012 [27] was followed in accordance with the ethical standards outlined by INIFAP. Faecal material was used to elaborate coprocultures and incubated for 7 days until the third larval stage was obtained. Larvae were recovered using a Baermann funnel system and washed via differential centrifugation using 40% sucrose for 3–5 min. Larvae were rinsed with tap water several times to eliminate sucrose residues. Subsequently, the larvae were unsheathed with 0.187% sodium hypochlorite solution for 3–5 min and washed again to discard the sheaths [28]. Clean larvae were resuspended in sterile distilled water and immediately used to perform an in vitro larval mortality assay. We used unsheathed *H*. *contortus* infective larvae because these larvae possess a high similarity with the histotrophic larval stage (or fourth larval stage) in terms of their cuticular structural characteristics.

### 2.7. Flavocillium subprimulinum Culture Filtrates

Liquid culture media were prepared using organic sweet potato dextrose broth (SPDB), potato dextrose broth (PDB), and Czapek–Dox broth (Cz-DoxB) (DIBICO, Stamford, CT, USA). The SPDB and PDB media were prepared using 200 g of either sweet potato or potato and 20 g of dextrose (Merck KGaA, Darmstadt, Germany). Sweet potatoes were boiled with 800 mL of distilled water for 25 min and sieved through gauze to separate the solid material, retaining only the liquid phase. The resulting liquid was spiked with 20 g of dextrose, and the mixture was brought to a volume of 1 L. Cz-DoxB was prepared using 35 g of medium dissolved in 1 L of distilled water, as recommended by the manufacturer. Subsequently, the media were distributed in 250 mL volume flasks containing 50 mL each and sterilised in an autoclave for 25 min. Sterile liquid media were inoculated with three cylindrical plugs (1 cm height × 1 cm diameter) from the agar surface of the fungus. The fungi in the flasks were incubated in the three media for 21 days at 18–25 °C. After the incubation period, the fungal mycelia were separated as follows: fungi and their liquid cultures were passed through five different filter systems as follows: (1) conventional coffee filter paper (ULINE, Apodaca, Mexico); (2) Whatman Paper No. 4 (Merck KGaA, Darmstadt, Germany); (3) glass fibre filter with binder of a 2 µM membrane micro-filter (Millipore, Rahway, NJ, USA); (4 and 5) nitrocellulose membrane filter discs of 0.45 and 0.22 µM, respectively (Millipore, Merck KGaA, Darmstadt, Germany). The obtained material was concentrated using a rotatory evaporator (Büchi R-300, Flawil, Switzerland) and completely dried via lyophilisation (Labconco, Kansas, MO, USA).

### 2.8. Assessment of the Lethal Activity of Fungal Liquid Culture Filtrates against Haemonchus contortus Infective Larvae

The dried fungal liquid culture filtrate (FLCF) was finally re-constituted by adding PBS (pH 7.2) according to the required concentration. The FLCF/nematode interaction evaluation was performed using 96-well microtiter plates (Costar, Cambridge, MA, USA). Then, 50 µL of FLCF were deposited in each well (n = 4), along with 50 µL of an aqueous suspension containing 100 *H*. *contortus* L3 (n = 3). The negative controls were as follows: (1) PBS, pH 7.2; (2) dried Cz-DoxB medium; (3) dried PDB medium; and (4) SPDB dried medium (2, 3, and 4 were reconstituted in PBS, pH 7.2). The following three FLCF concentrations were assessed: 100, 50, and 25 mg/mL. Readings were recorded 48 h post-treatment. Both motionless and moving larvae were observed and quantified under a microscope using 5× and 10× objectives. The differentiation between live and dead larvae was based on the following characteristics: motionless larvae were considered dead when, after touching their cuticle with a metallic fine needle, they remained motionless. In contrast, motionless larvae that moved after applying this physical stimulus were considered live larvae. The means of dead and live larvae were recorded and compared among the experimental groups [29]. The whole experiment was repeated three times.

### 2.9. Protein Profile and Protease Activity Analysis Using Liquid Culture Filtrates

Protein profile analysis was performed using lyophilised liquid culture filtrates. The lyophilised FLCF was resuspended in PBS (pH = 7.2) at 100 mg/mL, and the total protein concentration was estimated using a Qubit™ Protein Assay Kit (Invitrogen, Eugene, OR, USA). The proteins from the fungus produced in each culture medium were separated for their observation by sodium dodecyl sulphate–polyacrylamide gel electrophoresis (SDS–PAGE) at 5% and 12%. In addition, zymogram gel electrophoresis (5% gelatine) was performed to visualise the protease activity. A total of 3 µg of total protein were used for SDS-PAGE, and 1 µg was added to the zymogram gel for each treatment. The treatments evaluated in both gels were established as follows: PDB with fungus, PDB without fungus, SPDB with fungus, SPDB without fungus, Cz-DoxB with fungus, and Cz-DoxB without fungus [30].

### 2.10. Microscopic Analysis of Haemonchus contortus Infective Larvae Exposed to Flavocillium subprimulinum Liquid Culture Filtrates

The changes observed in the morphology of *H*. *contortus* infective larvae after 48 h of exposure to the highest concentration (100 mg/mL) of the FLCF were photographed using a LEICA DM6 (Wetzlar, Germany).

### 2.11. Statistical Analysis

The results obtained from the nematocidal activity assessment against *H*. *contortus* L3 were analysed using an analysis of variance (ANOVA), followed by a multiple comparison of means using the Tukey method (α = 0.05). A logistic regression was performed using the PROBIT model, and lethal concentrations 50 and 90 (LC50 and LC90, respectively) were obtained for the three media.

## 3. Results

### 3.1. Fungal Traditional Taxonomy (Morphometrics)

Regarding the macroscopic characteristics of *F*. *subprimulinum* growing in PDA plates after 10 days, the growth appeared circular and concentric, with a 4.5 cm diameter and a whitish and cottony aspect from the front view; the reverse of the plate appeared yellowish (Figure 2A,B). In flasks with liquid media, the fungus covered almost the whole surface of the medium with whitish and cottony mycelia, forming wrinkles (Figure 2C–E). After 10 days of incubation, this isolate showed slow and poor growth in the PDA medium. In contrast, an abundant production of mycelial biomass was observed in the three liquid culture media (Cz-DoxB, SPDB, and PDB) (Figure 2C–E).

Regarding the microscopical analysis of the aerial structures, the isolated fungus showed the following characteristics: the presence of macroconidia and microconidia arising from the apex of the lanceolate phialides and the presence of verticillated conidiophores with one to three phialides (Figure 3).

Table 1 shows the results of the observation and measurement of the principal fungal structures. The morphological characteristics of our isolate, according to our analysis, led us to conclude that this fungus belongs to the species *F*. *subprimulinum*, although this classification was corroborated via molecular identification.

### 3.2. Fungal Molecular Taxonomic Identification

After sequencing, a similarity analysis was performed, resulting in a range from 98.8 to 99.3% for the TEF1-α region and 99.3 to 100% for the ITS region and query covers from 98 to 99% for TEF1-α and 96 to 98% for ITS. Alignment analysis indicated that our isolate was closer to *F*. *subprimulinum*. Table 2 shows the five strains with the highest query covers and similarity percentages found in the NCBI database for both regions employed.

The phylogenetic trees of our *Flavocillium subprimulinum* isolate generated with 37 sequences for ITS and TEF1-α regions reported in the NCBI database are shown in Figure 4.

### 3.3. Assessment of the Lethal Activity of Fungal Liquid Culture Filtrates against Haemonchus contortus Infective Larvae

The mortality percentages of *H*. *contortus* infective larvae after 48 h of exposure to FLCF at the three concentrations in PDB, SPDB, and CzDoxB are shown in Table 3.

The results of the logistic regression by PROBIT are shown in Figure 5. The LC50 and LC90 values were estimated for each treatment.

### 3.4. Microscopic Analysis of Haemonchus contortus Infective Larvae Exposed to Flavocillium subprimulinum Liquid Culture Filtrates

A set of microphotographs showing the changes observed in the larvae exposed to the FLCF is shown in Figure 6.

Most larvae exposed to FLCF from Cz-DoxB appeared stretched and lost the normal architecture of their intestinal cells. Additionally, some structures with granules or, potentially, crystals were seen in the larvae intestine (Figure 6B(a)). Interestingly, the external cuticle coat appeared smooth, without any apparent changes (Figure 6B(b)). Larvae exposed to FLCF from PDB appeared slightly curved, and a similar aspect was observed for larvae exposed to FLCF from Cz-DoxB, with a loss of the integrity of gut cells (Figure 6C(a)). However, no intestinal granules were found (Figure 6C). In contrast, in larvae exposed to FLCF obtained from SPDB, no apparent change was observed, either in the intestinal cells or in the cuticular surface coat, which appeared smooth and comparable to that of unexposed larvae from the control group (Figure 6D(a,b)).

### 3.5. Protein Profile and Protease Activity Analysis from Liquid Culture Filtrates

Protein profile analysis of the FLCF is shown in Figure 7. The following protein bands were visualised using SDS-PAGE of *F*. *subprimulinum* (STV-INIFAP F. sub-strain) grown in the three-culture media: in SPDB, four protein bands weighing 90, 40, 25, and 23 kDa; in PDB, five protein bands weighing 150, 55, 40, 25, and 20 kDa; and in CzDoxB, five protein bands weighing 150, 55, 40, 30, and 17 kDa. No protein bands were observed in the lines with the three media without any fungus (controls).

The zymogram of *F*. *subprimulinum* grown in three liquid media and controls is shown in Figure 8. The zymogram revealed the presence of two bands with protease activity, weighing 72 and 14 kDa, in the fungus grown in the SPDB medium. In PDB, three bands with protease activity were identified: 40, 14, and 10 kDa. In CzDoxB, four protein bands with protease activity were identified, weighing ~95, 55, 40, and 30 kDa. The control lines with only media without any fungi were negative.

## 4. Discussion

### 4.1. Fungal Traditional and Molecular Taxonomy

Macroscopic analysis of the plates revealed growth with cottony and light, yellowish aspects in the reverse of the plates. Likewise, the microscopic analysis showed the presence of macroconidia and microconidia and lanceolate phialides, among other characteristics. These characteristics were similar to those of the three species belonging to the genus *Flavocillium*: *Flavocillium bifurcatum*, *F*. *primulinum,* and *F*. *subprimulinum*. Table 4 shows the similarities and differences between the species reported by Huang et al. [12] and Wang et al. [13] and our isolate.

Interestingly, 100% and 99% similarities were found with *F*. *bifurcatum* in the sequences obtained with the ITS and TEF1-α regions, respectively. However, the highest query covers (98%) and similarities (99.3%) were found in relation to *F*. *subprimulinum* (accession numbers: MK579178.1 for ITS and MG585321.1, MG585317.1 for TEF1-α). In addition, some morphological differences were found between our isolate and *F*. *bifurcatum* isolated from the larvae of Lepidoptera: Noctuidae. Previously, the species *F*. *subprimulinum* was isolated from soil, such as our isolate. Additionally, the macroconidia were smaller in *F*. *bifurcatum* (5.5–9.2 µm) [13] than in *F*. *subprimulinum* (7–14 µm) [12]. There is an additional finding that is unusual in relation to the species reported, namely the presence of a septum in the middle macroconidia, which was not identified in any of the species previously reported. We assume that this characteristic indicates that this isolate could be a sub-species. However, at this time, this is only a hypothesis, and further studies are needed to confirm it. Additionally, the constructed phylogenetic tree, using the conserved regions ITS and TEF1-α, shows that our isolate has a common ancestor with *F*. *subprimulinum* (posterior probability = 0.984). This finding helped us corroborate the morphological taxonomical identification.

### 4.2. Assessment of the Lethal Activity of Fungal Liquid Culture Filtrates against Haemonchus contortus Infective Larvae

The results of the exposure of *H*. *contortus* infective larvae to the FLCF of *F*. *subprimulinum* showed important activity in the three assessed media. Interestingly, the FLCF obtained from PDB and CzDoxB media showed a high mortality from 50 mg/mL, reaching values close to 75 and 78%, respectively. Notably, the FLCF obtained from CzDoxB showed a mortality value close to 60% at a concentration of only 25 mg/mL. The highest larval mortality was recorded with FLCF produced in PDB and CzDoxB at 100 mg/mL, reaching 95 and 96%, respectively.

### 4.3. Microscopic Analysis of Haemonchus contortus Infective Larvae Exposed to Flavocillium subprimulinum Liquid Culture Filtrates

The morphological changes observed in larvae after exposure to FLCF were interesting, such as the loss of the intestinal cell architecture observed in larvae exposed to FLCF obtained from the fungus cultured in Cz-DoxB and PDB. Although there are some reports on the nematocidal activity of many plants and plant extracts, there is little information about the nematocidal activity produced by myco-constituents from nematophagous fungi grown in liquid culture filtrates. In a recent study, a similar change regarding the loss of intestinal cell organisation was observed in *H*. *contortus* larvae exposed to the FLCF of the ascomycete *Arthrobotrys musiformis* after growing in Cz-DoxB [31]. A similar morphological alteration has been observed in larvae exposed to isorhamnetin, a flavonoid compound obtained from the leaves of *Prosopis laevigata*, a tree species used in traditional medicine [32,33]. Regarding the appearance of granules in the intestines of *H*. *contortus* infective larvae after exposure to the LCF obtained from Cz-DoxB, the authors of the present study do not have a convincing explanation for this fact. The use of confocal laser scanning microscopy analysis may be helpful in demonstrating a co-localisation effect of bioactive compounds into larval tissues to determine the mechanism underlying a possible mode of action [34]. A list of secondary metabolites produced by nematophagous fungi with nematocidal activity against different nematodes, bacteria, and fungi of agriculture and livestock importance is shown in Table 5.

Despite an extensive literature review, only a few reports of the nematocidal activity of nematophagous fungi liquid culture filtrates against nematodes of importance, either in agriculture or in the livestock industry, were found. In some studies, focused on Hypocreales, several bioactive compounds have been identified, e.g., saponins (glycosides), coumarins (phenolic compounds), and compounds related to farnesylated cyclohexenoxides from the oligosporon group, such as oligosporones and flagranones, with an important nematocidal activity [45,46]. A list of fungi belonging to the Orbiliales group with important nematocidal activities has been published by Degenkolb and Vilcinskas [47]. Future studies will need to identify the nematocidal myco-compounds produced by *F*. *subprimulinum*. In a recent study, the hypocreal fungus *Trichoderma asperellum* FbMi6 showed substantial nematocidal activity under laboratory conditions against the root-knot nematode *Meloidogyne incognita*, with egg-hatch suppression (96.6%) and juvenile mortality (90.3%) of the parasitic nematode. A high flavonoid content has been found in neem cakes enriched with this fungus [44]. These genera/species of fungi may be of great interest regarding their potential use against *H*. *contortus*, which is considered the most pathogenic parasitic nematode in cattle and small ruminants worldwide.

### 4.4. Protein Profile and Protease Activity Analysis of Fungal Liquid Culture Filtrates

SDS-PAGE revealed that FLCF from *F*. *subprimulinum* grown in three different culture media showed different protein profiles, although one protein band weighing ~40 kDa was expressed in the three media. Likewise, another band weighing ~25 kDa was visualised in both SPDB and PDB. Additionally, two more bands, weighing ~55 and ~150 kDa, were expressed in PDB and CzDoxB. Interestingly, in each medium, different protein bands were observed using SDS-PAGE, which were not expressed in the other media. Two bands of ~23 and ~90 kDa were expressed only in SPDB, one band of ~20 kDa was only expressed in PDB, and two bands of ~17 and ~30 kDa were only seen in CzDoxB. These differences observed in the protein band profiles in the different FLCFs assessed clearly show that the medium induces a different protein pattern, although some proteins were expressed in the different media and not influenced by the culture medium.

The protein band of ~40 kDa, which was observed in the protein profile in SDS-PAGE, was also observed in the zymogram in PDB and CzDoxB media, with important protease activity. Interestingly, this band was not visualised in the PDB. Three bands of ~30, ~40, and ~55 kDa, which were observed in CzDoxB in the protein profile, were also found in the zymogram, with an important protease activity. Two bands of ~14 and ~72 kDa were expressed in the zymogram in SPDB, but they were not observed in the protein profile. Likewise, another band of ~95 kDa was not identified in the protein profile. However, this band was expressed in the zymogram with protease activity in PDB and CzDoxB. Additionally, a band with only ~10 kDa, with protease activity, was observed in PDB, but it was not visualised in the protein profile. The differences found between the SDS-PAGE technique and the zymogram could be due to the experimental conditions. Nevertheless, the bands expressed in the zymogram showed clear evidence that the fungus *F*. *subprimulinum* produces proteins with protease activity in the three assessed media. Another interesting issue is that the FLCFs were processed by rotavapor and freezing before being used in the protein and zymogram assessment. These processes can affect protein integrity. However, the zymogram showed clear protease activity of a number of *F*. *subprimulinum* proteins secreted in the three media. Another important issue is that the control lines in the gel were free of protein bands, supporting our results. Many protein bands with protease activity have been reported for different nematophagous fungi, and this activity has been associated with the pathogenicity of fungi against nematodes (Table 6). It is important to consider that the nematode cuticle is constituted by proteins, saccharides, and lipids [48], among other tissue components.

The invasion of nematophagous fungi to nematodes is mediated by mechanical and biochemical processes in which proteases play an important role in the invasion and degradation of nematode tissues [16]. The present study is the first report of the presence of verticillated microfungus *F*. *subprimulinum* (Hypocreales) in Mexico. In the present study, we found protease activity in the LCF of *F*. *subprimulinum*, which could be responsible for the lethal effect of these filtrates. Nevertheless, we do not discard the possibility of secondary metabolites with nematocidal activity in LCF. Thus, we will continue to work with the filtrates of this fungus to explore the possibility of finding potential compounds derived from the secondary metabolism of the fungus with nematocidal activity. The results represent a basis for future studies to explore the potential use of this fungus against the blood-feeding nematode *H*. *contortus*, the main pathogenic parasite affecting small ruminants all over the world.

## 5. Conclusions

In this study, an isolate of the Hypocreales fungus of the species *F*. *subprimulinum* was isolated for the first time in Mexico. This isolate was taxonomically characterised and the fungus was cultured in different liquid media. The FLCF were assessed to investigate their potential in vitro nematocidal activity; the FLCF of this isolate cultured in CzDoxB showed the highest nematocidal activity of 95.82% against infective larvae of the blood-feeding nematode *H*. *contortus*.

## Figures and Tables

**Figure 1 jof-10-00056-f001:**
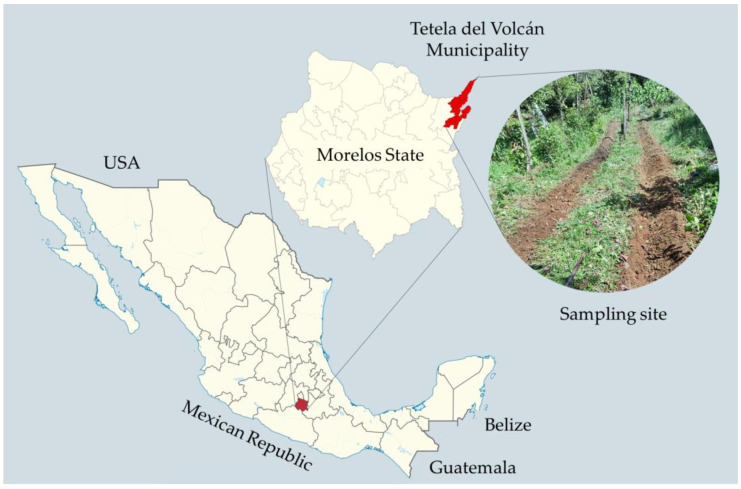
Map of the Mexican Republic, the State of Morelos, the Tetela del Volcán Municipality, and a picture of the sample site.

**Figure 2 jof-10-00056-f002:**
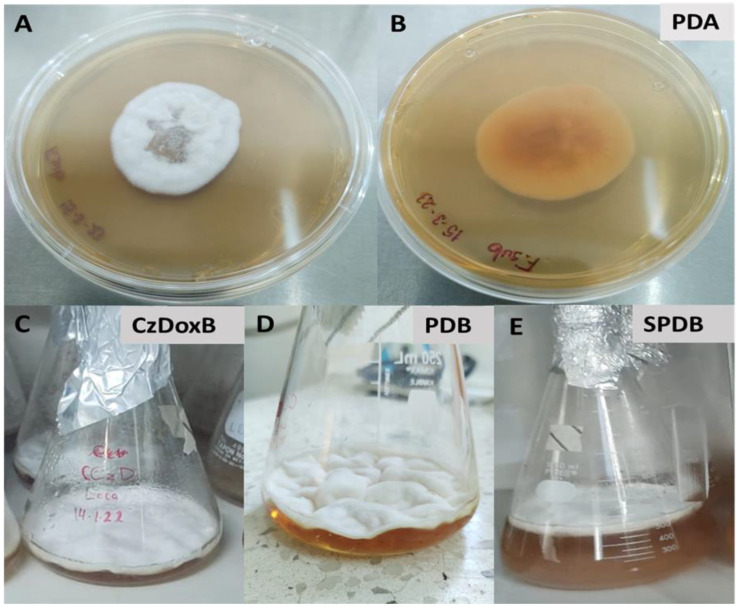
Aspect of *Flavocillium subprimulinum* (STV-INIFAP F. sub-strain) after 10 days of growth on Petri dishes with potato dextrose agar (PDA) (**A**,**B**) and in flasks containing three different liquid culture media: (**C**) Czapek–Dox broth, (**D**) potato dextrose broth, and (**E**) sweet potato dextrose broth.

**Figure 3 jof-10-00056-f003:**
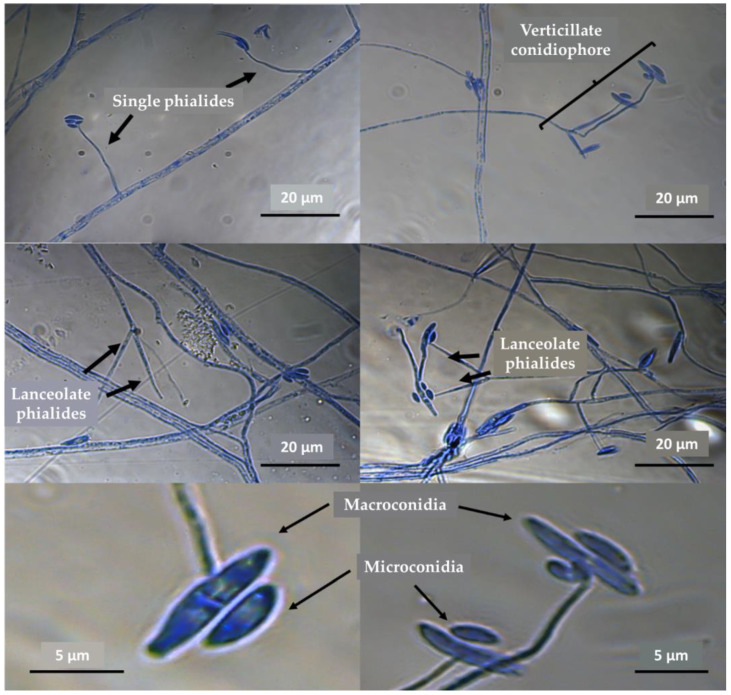
Microphotographs showing the aspects of macroconidia and microconidia, singe and lanceolate phialides and verticillate conidiophores of the fungus *Flavocillium subprimulinum* (STV-INIFAP F. sub strain).

**Figure 4 jof-10-00056-f004:**
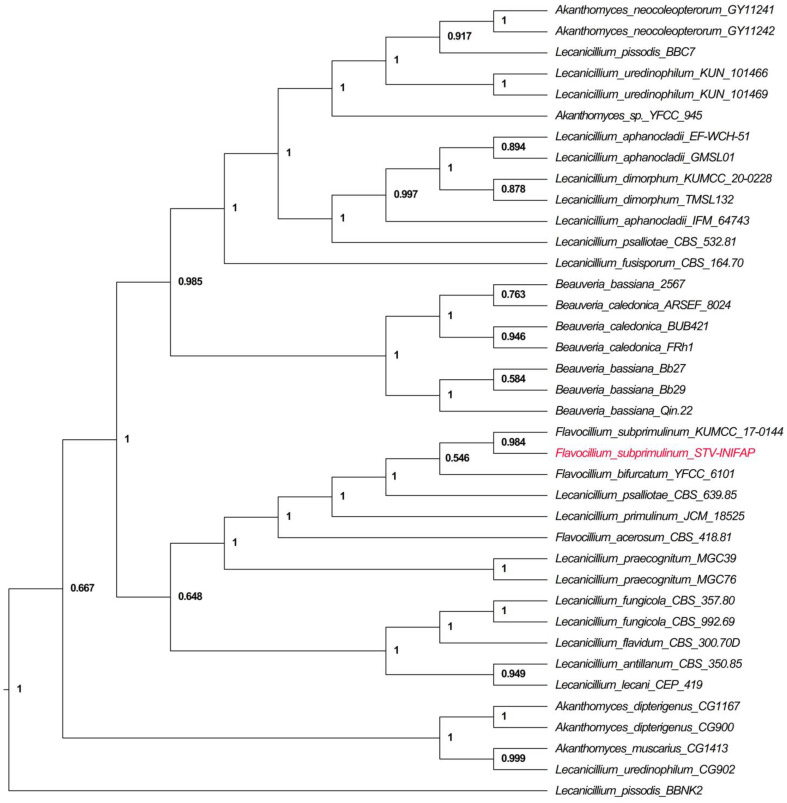
Phylogenetic tree constructed by Bayesian Inference with the ITS-TEF-linked sequences using sequences from strains reported in the NCBI database to compare with *Flavocillium subprimulinum* (STV-INIFAP F. sub-strain, highlighted in red color). Note: Values in the nodes are the posterior probabilities; only values above 0.5 were considered.

**Figure 5 jof-10-00056-f005:**
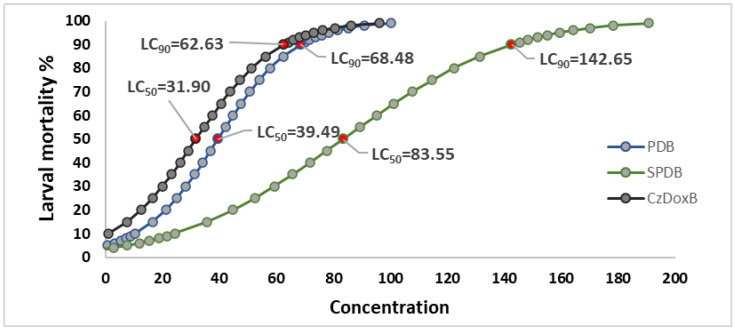
Logistic regression by the PROBIT method; lethal concentrations 50 and 90 (LC50 and LC90) estimated for the three media culture filtrates, potato-dextrose broth (PDB), sweet potato-dextrose broth (SPDB), and Czapek–Dox broth (CzDoxB). R^2^ = 0.92.

**Figure 6 jof-10-00056-f006:**
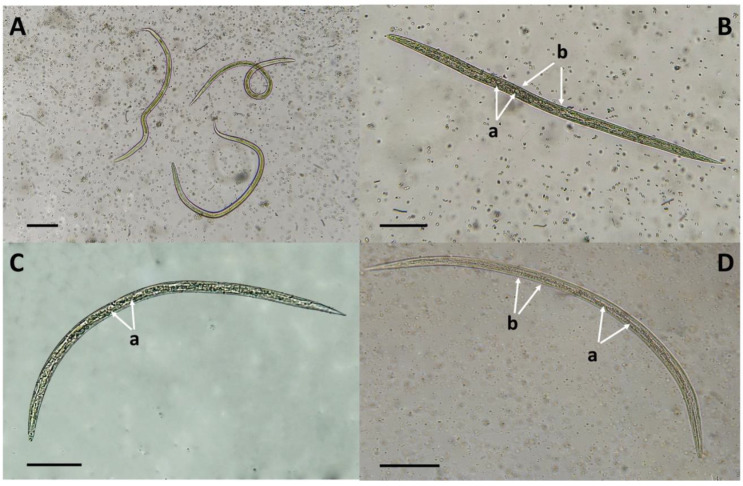
Aspects of *Haemonchus contortus* unsheathed infective larvae after 48 h of exposure to *Flavocillium subprimulinum* liquid culture filtrates (FLCF). (**A**) Live larvae (from the control group); (**B**–**D**) Larvae exposed to FLCF from Czapek–Dox broth, potato dextrose broth, and sweet potato dextrose broth, respectively. (Ba) Granules or crystals into the intestine; (Bb) smooth external cuticle without any apparent change; (Ca) loss of the integrity of gut cells; (Da,b) no apparent changes in intestinal cells or at the cuticle. Bar scale = 100 µm.

**Figure 7 jof-10-00056-f007:**
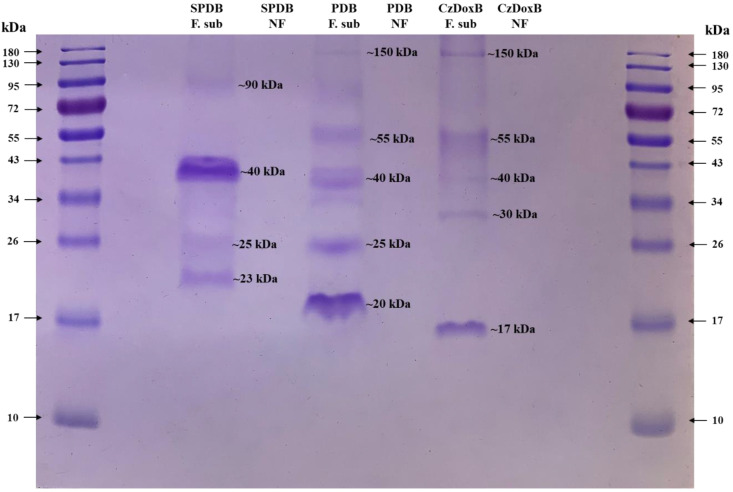
*Flavocillium subprimulinum*-secreted proteins visualised using SDS-PAGE. Lanes 1 and 10: molecular weight; Lanes 3, 5, and 7: *F*. *subprimulinum* cultured in sweet potato dextrose broth (SPDB), potato dextrose broth (PDB), and Czapek–Dox broth (CzDoxB), respectively; Lanes 4, 6, and 8: SPDB, PDB, and CzDoxB medium alone (without fungus), respectively.

**Figure 8 jof-10-00056-f008:**
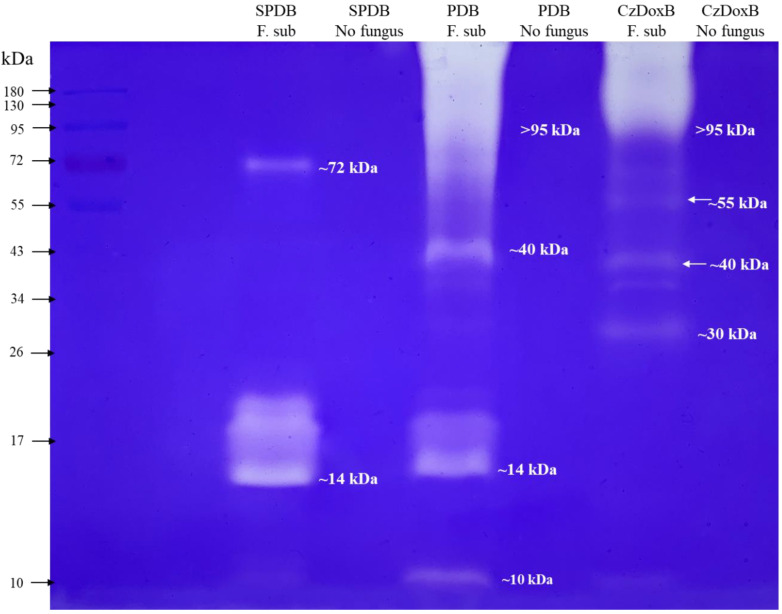
Zymogram of *Flavocillium subprimulinum* (STV-INIFAP F. sub-strain) grown in three liquid media, sweet potato dextrose broth (SPDB), potato dextrose broth (PDB), and Czapek–Dox broth (CzDoxB), as well as their controls without fungus (Controls).

**Table 1 jof-10-00056-t001:** Means and ranges of the principal morphological characteristics obtained from the *Flavocillium subprimulinum* isolate (STV-INIFAP F. sub strain).

Characteristic	n	Length (µm)	Width (µm)
Macroconidia	50	9.22 (7–14)	2.11 (1.5–2.6)
Microconidia	50	4.46 (3–7)	1.81 (1.2–2.4)
Phialides	50	26.05 (14–45)	1.52 (1–2.5)

**Table 2 jof-10-00056-t002:** List of the highest percentages of coverage and similarity for *Flavocillium* and *Lecanicillium* species corresponding to both sequenced regions (ITS and TEF1-α), as well as the accession number of strains reported at the NCBI Database.

Strain	Query Cover %	Similarity %	Gen Bank Accession Number
ITS			
*Flavocillium subprimulinum*	98	99.3	MK579178.1
*Lecanicillium* sp.	98	99.3	KX496884.1
*F. bifurcatum*	96	100	NR_173888.1
*F. bifurcatum*	96	100	MN576834.1
*F. bifurcatum*	96	100	MN576833.1
TEF1-α			
*F. subprimulinum*	98	99.3	MG585321.1
*F. subprimulinum*	98	99.3	MG585317.1
*F. bifurcatum*	99	99.0	MN576951.1
*Lecanicillium* sp.	98	99.0	KM283809.1
*Lecanicillium* sp.	98	98.8	KM283824.1

**Table 3 jof-10-00056-t003:** Mortality rates obtained for each concentration in three fungal liquid culture filtrates of *Flavocillium subprimulinum* (STV-INIFAP-F. sub strain): potato dextrose broth (PDB), sweet potato dextrose broth (SPDB), and Czapek–Dox broth (CzDoxB).

Concentration mg/mL	Mortality Percentage (Mean ± SE)		
	PDB	SPDB	CzDoxB
0	2.41 ± 1.48 a	2.17 ± 1.69 a	3.72 ± 2.58 a *
25	29.89 ± 6.32 b	16.07 ± 11.79 c	58.44 ± 8.10 a
50	74.95 ± 9.92 a	21.01 ± 5.27 b	77.99 ± 8.46 a
100	94.43 ± 2.42 a	64.00 ± 8.97 b	95.82 ± 3.50 a

* Means with different letters at the same concentration are significantly different (*p* < 0.05).

**Table 4 jof-10-00056-t004:** The main characteristics of three similar fungal species of the genus *Flavocillium* compared with the species isolated from Tetela del Volcán Municipality, Morelos, Mexico, in the present study.

Characteristics	*Flavocillium bifurcatum* ^(1)^	*F*. *primulinum* ^(1)^	*F*. *subprimulinum* ^(1,2)^	*F*. *subprimulinum* (Present Study)
Isolation source	Larvae of Noctuidae (Insecta)	Soil	Soil	Agricultural soil
Macroconidia characteristics	Cymbiform	Cylindrical to ellipsoidal Single-celled	Obovoidal to ellipsoidal, elongated, straight or slightly curved, non-septate, single, or usually aggregated at the apex of phialides	Cylindrical to ellipsoidal, slightly curved, single or in groups, one for each phialide. Aseptate or occasionally one-septate.
Measurements	5.5–9.2 × 1.3–2.7 μm	3–9.5 × 1–2.5 μm	4–15 × 2–6 μm	7–14 × 1.5–2.6 μm
Microconidia characteristics	Ellipsoidal to reniform	Oval to ellipsoidal, aggregated in apical heads.	Oval to ellipsoidal	Oval to ellipsoidal or globose, aggregated at the apex of phialides, 1 or up to 4 in each phialide
Measurements	2.1–4.2 × 0.9–1.5 μm	NA	NA	3–7 × 1.2–2.4 μm
Colony	White to yellowish, cottony with a raised mycelial density at the centrum, generating several concentric rings at the edge, reverse pale yellow to brown	White to yellowish	White, usually raised dome-shaped mycelium, dense with a sunken zone at the centrum, verrucose around the margin. Slow growth in PDA	Slow growth in PDA, fast growth in liquid cultures such as Czapek Dox broth or sweet potato dextrose broth. White, circular, cottony, slightly yellowish reverse
Hyphae	Hyaline, septate, branched, smooth-walled, 1–2.3 µm, conidiophore measuring 50–64.2 × 0.9–1.8 µm	Hyaline, septate	Hyaline, branched and septate, from 1–3 µm, conidiophore measuring 19–32 × 1.5–3.5 µm (24 × 2.5 µm). Phialides solitary or up to three at the node	Hyaline, branched, 1–3 µm diameter, conidiophores with one or up to three phialides.
Phialides	Lanceolate, single, or in whorls of two to five from 18.1–44.5 µm × 1.1–24 µm in size	Produced on prostrate aerial hyphae, solitary or in whorls of 2–5 that taper toward the apex	Lanceolate, occurring directly from the prostrate hyphae, solitary or two to three phialides, gradually attenuated toward the apex	Lanceolate, arising from aerial hyphae, wide at the base, narrowed at the apex. Measuring 14–45 × 1–2.5 µm

^(1)^ Wang et al. [13]; ^(2)^ Huang et al. [12]; NA = Not Available.

**Table 5 jof-10-00056-t005:** Nematophagous fungi liquid culture filtrates assessed against different targets, including nematodes, bacteria, and fungi.

Fungus	Target	Activity	Reference
*Purpureocillium lilacinum*	*Meloidogyne* spp. *Heterodera rostochiensis* *Apelenchoides* spp. *Neoplectana* spp.	Nematocidal	[35]
*P. lilacinum* *Trichoderma* *longibrachiatum*	*Meloidogyne* sp. *Heterodera* sp. *Radopholus* sp. *Pratylenchus* sp.	Nematocidal	[36]
*Verticillium leptobactrum*	*Meloidogyne incognita*	Nematocidal	[37]
*Arthrobotrys dactyloides* *A. oligospora* *Dactylella brochophaga*	*Escherichia coli* *Staphylococus aureus*	Antibacterial	[38]
*T. harzarium*	*Meloidogyne* spp.	Ovicidal and nematocidal	[39]
*A. oligospora*	*Caenorhabditis elegans*	Nematocidal	[40]
*A. musiformis*	Gastrointestinal nematodes	Nematocidal	[41]
*T. hamatum*	*Acidovorax avenae* *Alternaria radicina* *M. incognita* eggs	Antibacterial Antifungal Nematocidal	[42]
*A. musiformis*	*Haemonchus contortus*	Nematocidal	[43]
*T. asperellum*	*Meloidogyne incognita*	Nematocidal	[44]

**Table 6 jof-10-00056-t006:** Biological activities of different nematophagous fungal proteins and nematode targets.

Fungus	Protein	Molecular Weight (kDa)	Target	Reference
*Trichoderma pseudokoningii*	Serine protease SprT	31	*Meloidogyne incognita*	[49]
*Verticcillium chlamydosporium*	Chymoelastase-like protease VCP1	33	*Meloidogyne incognita*	[50]
*V. suchlasporium* *V. chlamydosporium*	Endochitinase CHI43 Serine protease P32	43 32	*Globodera pallida*	[51]
*Lecanicillium* sp.	Endochitinase Choline dehydrogenase	37 68	*Meloidogyne javanica* eggs	[52]
*Arthrobotrys conoides*	Glucoamylase GAA	83–87	NA	[53]
*Pochonia chlamydosporia and P. rubescens*	Serine protease P32	32	NA	[54]
*P. lilacinum* *L. psalliotae*	Ver 112 PL 646	32	*C. elegans*	[55]
*Dactylella shizishanna*	Serine protease	35	*Panagrellus redivivus*	[56]
*Monacrosporium sinense*	Proteases Ms1, Ms2 and Ms3	37 33 11	*P. redivivus*	[57]
*Hirsutella rhossiliensis*	Alcaline protease Hasp	33	*Heterodera glycines*	[58]

## Data Availability

Data are contained within the article.

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
