# Peer review of "First Record of Flavocillium subprimulinum (Cordycipitaceae, Hypocreales) in Mexico: Morphological and Molecular Characterisation, Nematocidal Activity of Its Liquid Culture Filtrates against Haemonchus contortus and Protease Activity"

_jof, 2024, doi:10.3390/jof10010056_

Round 1
Reviewer 1 Report (Previous Reviewer 2)
Comments and Suggestions for Authors
The authors explained that in their response that exsheathed larvae were used to mimic a parasitic stage. This is an important point, since their target is not infective larvae in the field but rather parasitic stages and thus rather an anthelmintic activity. I did not see anything about it in the modified version. It should be mentioned either in the introduction or at least in the materials and methods.
Author Response
Reviewer 1 comment
The authors explained that in their response that exsheathed larvae were used to mimic a parasitic stage. This is an important point, since their target is not infective larvae in the field but rather parasitic stages and thus rather an anthelmintic activity. I did not see anything about it in the modified version. It should be mentioned either in the introduction or at least in the materials and methods.
Author´s response:
A brief paragraph about this important issue has been inserted in Page 4, lines 177-179.

Reviewer 2 Report (New Reviewer)
Comments and Suggestions for Authors
This study obtained an isolate of Flavocillium subprimulinum from Mexico, and they determined the nematocidal activity of it culture filtrates based on three liquid media. Here, I have the following main questions:
1. The ITS and TEF sequences of their isolate had high similarity with Flavocillium subprimulinum and F. bifurcatum (both >99%). Their ITS tree is not accordant with TEF tree, so, they can construct phylogenetic tree combined ITS and TEF sequences and just choose these sequences from type strain of all Flavocillium species. Additionally, the microcosmic morphological feature of their isolate has difference, such the conidia Flavocillium subprimulinum described by HUANG et al. is aseptate and 2-6 µm ( mean 4 µm) wide while this study observed 1-septate and 1.5-2.6 µm wide macroconidia. Their should further determine the taxonomy of the isolate. if possible, they can amplify other gene (e.g., nrLSU, nrSSU, rpb1 or rpb2) to construct tree.
2. This study just do protein profile analysis, not include protease activity.
3.More comments could be explored in the manuscript.

Comments on the Quality of English LanguageExtensive editing of English language required.
The description of methods and results should be more specific and accurate.
Author Response
Reviewer 2 Comments
This study obtained an isolate of Flavocillium subprimulinum from Mexico, and they determined the nematocidal activity of it culture filtrates based on three liquid media. Here, I have the following main questions:
The ITS and TEF sequences of their isolate had high similarity with Flavocillium subprimulinum and F. bifurcatum (both >99%). Their ITS tree is not accordant with TEF tree, so, they can construct phylogenetic tree combined ITS and TEF sequences and just choose these sequences from type strain of all Flavocillium species. Additionally, the microcosmic morphological feature of their isolate has difference, such the conidia Flavocillium subprimulinum described by HUANG et al. is aseptate and 2-6 µm ( mean 4 µm) wide while this study observed 1-septate and 1.5-2.6 µm wide macroconidia. Their should further determine the taxonomy of the isolate. if possible, they can amplify other gene (e.g., nrLSU, nrSSU, rpb1 or rpb2) to construct tree.
Author´s response:
Authors generated a combined tree based on ITS and TEF sequences to get a more robust phylogenetic tree. After analyzing this new tree, authors found a more reliable and precise molecular identification of this genus/species.
- This study just do protein profile analysis, not include protease activity.
Author´s response: Authors included a zymogram analysis showed in Figure 8.
3.More comments could be explored in the manuscript.
Author´s response: Everyone of the whole suggestions kindly made by Reviewer 2 on the PDF version (peer-review-33759164.v2.pdf) of our manuscript were attended.
Comments on the Quality of English Language
Extensive editing of English language required.
Author´s response: The manuscript was sent to a company specially focused on improvement of the English language
The description of methods and results should be more specific and accurate.
Author´s response: After the whole changes made to the new version of the manuscript, authors hope the paper has achieved a clearer scientific description.
Authors hope that after these changes, reviewers consider this manuscript with the excellent quality that characterize your prestigious journal,

Round 2
Reviewer 2 Report (New Reviewer)
Comments and Suggestions for Authors
Please check the manuscript again carefully.
E.g., line152 There are spaces in the head of the sentence.
line 572 BMC:Microbiology delete :
Author Response
Jiutepec City, Mexico 5th of January 2024
JOURNAL OF FUNGI
Editor
Dear Editor,
In relation with the manuscript entitled:
“First record of Flavocillium subprimulinum (Cordycipitaceae, Hypocreales) in Mexico: morphological and molecular characterisation, nematocidal activity of its liquid culture filtrates against Haemonchus contortus and protease activity”,
Whose authors are:
Gustavo Pérez-Anzúrez, Pedro Mendoza-de Gives, Agustín Olmedo-Juárez, María Eugenia López-Arellano, Génesis Andrea Bautista-García, Ana Yuridia Ocampo-Gutiérrez, Elke von Son-de Fernex, Miguel Ángel Alonso-Díaz, Edgar Jesús Delgado-Núñez and Adolfo Paz-Silva”
After 2nd round, me and my colleagues have revised and improved our manuscript according to the comments and suggestions provided by reviewer 2.
We have also prepared a list of answers and comments to the questions and recommendations provided by reviewers to our manuscript and the actions we took that we show it next:
Comments and Suggestions provided by Reviewer 2:
Please check the manuscript again carefully.
E.g., line152 There are spaces in the head of the sentence.
line 572 BMC:Microbiology delete :
Authors actions:
In attention to the comments provided by Reviewer 2, authors carefully revised the whole manuscript and we corrected every detail that we identified as wrong and we marked the corrected text in yellow color to facilitate the revision by the editors and the second reviewer.
Authors would be honored if after revision by the 2nd reviewer our manuscript satisfy the quality requirements that characterize your prestigious journal.
Thank you very much,
Dr Pedro Mendoza de Gives
Author for correspondence Researcher
Laboratory of Helminthology
National Center for Disciplinary Research in Animal Health and Innocuity INIFAP-Mexico, Ministry of Agriculture, Mexican Government.
Boulevard Paseo Cuauhnahuac No. 8534, Col. Progreso, Jiutepec, Morelos, México.

This manuscript is a resubmission of an earlier submission. The following is a list of the peer review reports and author responses from that submission.
Round 1
Reviewer 1 Report
Comments and Suggestions for Authors
This work is not yet ready to be published.
We are dealing with a manuscript submitted to the excellent journal “Journal of Fungi” and therefore we should pay attention to the high quality standard.
Below the authors will find some comments that will help to improve the article.
1. Introduction
Page 1, line 38 – text in different color
There is nothing in this section that introduces the reader to the knowledge of enzymes, specifically proteases, which were the subject of this article. Therefore, you need to add this information.
2. Materials and Methods
Page 2, line 74 - (Error! Reference source not found.) Please, correct it.
2.2.1 Fungal isolation - I suggest adding information on the exact sampling location, with latitude and longitude.
2.7 Flavocillium subprimulinum culture filtrates - The authors state that they concentrated the filtrate in a rotavapor. Next, the authors state that protease activity was verified using a zymogram. I wonder how the proteases were not denatured after the rotavapor concentration process? What temperature was used? What was the incubation time?
It is extremely difficult for the enzymes to have resisted such inhospitable conditions and IF they did, they were definitely not at 100% activity.
Why protease activity was not properly measured in catalytic units (U)? It is a simple test that would help to prove how much of the enzyme would actually still be active.
Why did the authors not demonstrate the nematophagous activity of the fungus?
Results
Page 8, line 248 - (Error! Reference source not found.) Please, correct it.
Discussion
The authors state that the nematicidal action of the filtrates is due to protease activity.
To assert this, authors should:
1) Have assembled a group with the filtrate with denatured enzymes. In this way, they would rule out the action of secondary metabolites;
2) Measured the protease activity of filtrates in order to prove and quantify the catalytic action of proteases;
3) Use appropriate culture media to induce protease production. The media used were very rich in carbohydrates, which inhibit the production of proteases. Furthermore, they are poor in N;
4) Do not use rotavaporization to concentrate the filtrates, denaturing most of the proteases.
Reviewer 2 Report
Comments and Suggestions for Authors
This is a documented work on the nematodicidal activity of the fungus Flavocillium subprimulinum first found in Mexico.
The first part of the paper is producing evidence that F. subprimulinum found in Mexico match with its description and molecular identification as published by Wang et al. in 2020. This is the case. (See errors indicated in table 2). However, the figures 4 and 5 do show that their substrain, is not so related to the other F. subprimulinum. Is there any explanation since the bootstraps indicate that this strangeness is present in 90 to 100 % of resamplings?
Most of hyphomycetes are predatory of nematodes by mechanical systems of traps. Is F. subprimulinum producing these traps like Arthrobotrys for example? If so, what is the interest of evaluating the toxic substances produced by F. subprimulinum? For the sake of explaining the toxic action of the fungus or to understand its complementary role of mechanical traps?
The authors have chosen to use larvae of the nematode Haemonchus contortus to measure its toxic/lethal effect. They use exsheathed larvae which are not the one to be in contact with the fungus in nature. Could they justify this choice?
The answers to these questions are needed before detailed comments can be properly made by the referee.